# Health Equity in Climate Change Policies and Public Health Policies Related to Climate Change: Protocol for a Systematic Review

**DOI:** 10.3390/ijerph19159126

**Published:** 2022-07-26

**Authors:** Sudeepa Khanal, Lira Ramadani, Melanie Boeckmann

**Affiliations:** 1Faculty of Health Sciences, University of Bielefeld, Universitätsstraße 25, 33615 Bielefeld, Germany or khanal.sudeepa@gmail.com (S.K.); l.ramadani@uni-bielefeld.de (L.R.); 2Faculty of Human and Health Sciences, University of Bremen, Mary-Somerville-Str. 3, 28359 Bremen, Germany

**Keywords:** policy analysis framework, policy analysis, health equity, social determinants of health, environmental equity, environment and public health, environmental policy, climate change policies, protocol

## Abstract

The relationship between risks to health equity and climate change is well established, and various international organizations and literature has called for intersectoral action for tackling these issues. While there is a growing recognition of the importance of equity-focused responses to climate change, if and how health equity measures are incorporated into climate-change-related policy decisions has not yet been adequately explored. There are numerous approaches and frameworks for conducting policy analysis, and for understanding equity in the context of climate change and public health separately; however, a comprehensive framework for exploring the complexity of these interacting factors is hard to find. This review aims to systematically identify policy analysis frameworks and studies focusing on health equity in climate change related policies. Five electronic databases will be searched for peer-reviewed articles in English and from 2000. Articles will be subjected to systematic retrieval and quality assessment, and thematic analysis will be used for data analysis. The study findings will provide insight into different existing policy-analysis frameworks and policy-analysis approaches to understand health equity considerations in climate change policies and in health policies related to climate change.

## 1. Introduction

Climate change is an urgent global issue affecting all countries [1]. The United Nations Secretary General, in 2018, called it the “major over-riding environmental issue of our time and the single greatest challenge facing environmental regulators” [2]. Climate change poses threats in various dimensions, and among its numerous impacts, its effect on human health has been a growing concern. The Intergovernmental Panel on Climate Change (IPCC) has acknowledged that climate change significantly contributes to the global burden of disease and premature deaths [3]. The latest report of IPCC also places human health and equity at the center for climate resilient development [4]. Similarly, the World Health Organization (WHO) estimates that an additional two hundred and fifty thousand deaths globally per year between 2030 and 2050 will be due to climate change [5]. The effects of climate change on public health occur both directly, due to extreme weather events, and indirectly, due to changes in environmental and socio-economic factors [6].

There is a global consensus that these climate-driven health risks and impacts are not evenly distributed across population groups, communities, or countries [7] and vulnerability to health issues due to climate change is determined by exposure, sensitivity, and adaptive capacity [8]. Climate change is expected to exacerbate health problems that already pose a major burden on vulnerable populations. Certain groups of the population have higher susceptibility to climate-sensitive health impacts owing to their age (children and elderly), gender (particularly pregnant women), social marginalization (associated in some areas with indigenous populations, poverty, or migration status), or other health conditions such as HIV, thus further amplifying inequities [9]. The need to address climate change and the equity issues it causes is of global importance and is evident by its being documented in the United Nations Framework Convention on Climate Change (UNFCCC), Kyoto Agreement in Paris and by the Sustainable Development Goals (SDGs) [9]. Already in 2008, the Commission on Social Determinants of Health (SDOH) of the WHO also proposed to bring the two agendas of health equity and climate change together. Likewise, various studies have also called upon the need to build strong coalitions addressing profound links between climate change and health equity [10].

Policies have powerful implications to achieve intended outcomes, and having equity-centered policy is considered a key to guide climate change plans and actions addressing the needs of the vulnerable population [11]. Given the growing recognition of the importance of equity-focused responses to climate change, empirical and policy studies to understand health equity in relation to the climate change have increased in the last decades. Nevertheless, health equity in the context of climate change often seems to be interpreted narrowly, and the existing literature in the area is dominated by studies on the effect of policies/specific interventions on the health outcomes or centered around the effects of climate change on health outcomes. These are particularly found in the forms of “health impact assessment (HIA)”, climate change vulnerability assessment, Environment Impact Assessment (EIA), and Social Impact Assessment (SIA), among others [12,13].

Policy analyses are critical to understanding the gaps in policies and are regularly undertaken to understand past policy successes and failures and to plan for future policy design and implementation [14]. Policy analysis frameworks are a key element of a credible policy analysis and involve the use of an analytical framework grounded in one or more theories and models of policy making. These frameworks help to identify and interpret the relationship between key variables relevant to the policy issue of interest [15]. Equity is an important dimension of health policy analysis, and there are many theories and frameworks in research where health equity is considered a central component of policy analysis. For example, John et al. (2014) have described examples of assessments of equity in policies as (a) procedural equity which assures that processes are in place and are fair and (b) substantive equity related to policies that help to reduce the disparity in outcomes among various populations [16]. Similarly, the Health Equity Measurement Framework (HEMF) is specifically designed to measure the direct and indirect effects of Social Determinants of Health (SDOH) to support improved statistical modelling and measurement of health equity [17]. There are also several Intersectionality-Based Policy Analysis (IBPA) Frameworks that offer ways to better understand the effects of policy on the diversity of populations in various health programs, including precisely identifying who is benefiting from and who is excluded from health policy goals [18].

While there is a large spectrum of methods specified in the literature for conducting a policy analysis focused on health equity, a systematic review by Mark G and Randall (2014) exploring the use of policy analysis theories or frameworks to understand health equity discusses how most of the policy analysis studies tend to focus on narrow aspect of health equity/SDOH rather than taking a holistic view. They suggest, by doing this there is the risk of under-emphasizing the role of other critical factors (such as social, economic, or political environments) for putting health equity issues on the policy agenda. They therefore call for the use of a more comprehensive application of theories and frameworks for policy analysis to produce more insightful analyses of health equity issues [19].

Just as there are a considerable number of frameworks for equity focused policy analysis in the health sector, several frameworks have also been developed for understanding equity and policy processes in the field of climate change. For example, based on the synthesis of various climate change literature, Reckien et al. (2017) have identified three types of equity: (a) outcome-based/distributive/consequential equity, relating to the consequences of a policy, action or developmental trend, (b) process-oriented/procedural equity, referring to impartiality and fairness in the process of delivering and administering justice, such as access to decision-making processes; and (c) contextual equity, linking the first two dimensions by taking into account pre-existing political, economic and social conditions [9]. The report by Rudolph L, Gould S, Berko et al. (2015) on Climate Change and Health (USA) [20] identifies policy and systems as one of the upstream interventions for far-reaching impacts to support health, equity, and sustainability in climate response. They state that policy and systems change spans many types of interventions, including changes in standards, regulations, and laws, funding distribution, financial and other incentives (such as subsidies, fees, or recognition awards), as well as changes in procurement and hiring processes. While many of these frameworks talk about equity, which also includes “health equity”, policy analysis frameworks specifically for understanding health equity in the climate change policy remain underexplored. Various organizations have also proposed different frameworks to facilitate the integration of these concepts at operational level. For example, Center of Disease Control, USA (CDC) has developed the “Building Resilience Against Climate Effects” (BRACE) framework, which proposes a five-step process for health officials to develop strategies and programs to help communities prepare for the health effects of climate change. These proposed steps are (a) anticipate climate impacts and assess vulnerabilities, (b) projection of the disease burden, (c) assessing public health interventions, (d) developing and adapting a climate and health action plan, and (e) evaluating impact and improving quality of activities [21]. Similarly, WHO has developed an operational framework for building climate-resilient health systems for clearly mainstreaming climate change into sector-wide or vertical programs. This framework proposes ten components covering areas of (a) leadership and governance, (b) health workforce, (c) vulnerability, capacity and adaptation assessments, (d) integrated risk monitoring and early warning, (e) health and climate research, (f) climate resilient and sustainable technologies and infrastructure, (g) management of environmental determinants of health, (h) climate informed health programs, (i) emergency preparedness and management and (j) climate and health financing. The framework further links these components with the WHO building blocks of health systems for focused and committed action for evidence informed policy making [22]. Similarly, WHO has also developed a guidance to ensure incorporation of health risk of climate change in the overall National Adaptation Planning (NAP) processes including considerations for equity and social determinants of health [23]. However, use of these proposed frameworks to understand health equity in climate change related policies in academic literature is unknown and remains to be explored.

In addition to the equity-focused policy analysis frameworks specific to public health and climate change sectors, a number of frameworks have also been developed to understand health equity in cross-cutting domains. One such example is the Health in All Policies (HiAP) framework, a collaborative approach that integrates and articulates health considerations into policymaking across sectors to improve the health of all communities and people [24,25]. Similarly, the health equity policy tool developed at WHO Europe intends to support WHO Member States and partners to strengthen the implementation of the principle of leaving no one behind principle and strategies through specific policy actions [26]. Meanwhile, CDC’s policy analytical framework expands on the domains of problem identification, policy analysis, and strategy and policy development [27]. In addition, many other frameworks exist, and application of these frameworks to understand health equity in the context of climate change is unknown. Moreover, to our knowledge there is no systematic account of the policy analysis studies conducted in the area. Therefore, this review aims to synthesize all pertinent frameworks for policy analysis used to understand health equity across the disciplines of climate change and public health related to climate change. The study will also provide a summary of various policy analysis studies used to understand health equity in the face of climate change. To our knowledge, the current study will be the first to synthesize the existing published literature on health equity focused policy analysis frameworks and map out various policy analysis studies specifically related to health equity and climate change policies.

### 1.1. Rationale

Health equity is increasingly identified as a principal goal to be achieved through policies and activities. Though the concept of health equity can mean different things from the perspectives of public health and climate change sectors [14], various studies have shown that the social determinants of health are the common underlying factors causing vulnerability to both health inequity and climate change effects on health, and tackling these requires intersectoral action [28]. Policy analysis is a complex process, and capturing various aspects of health equity in climate change related policies poses several challenges, including use of a comprehensive policy analysis framework with a holistic conceptualization of health equity [15]. There are numerous approaches and frameworks for conducting policy analysis, for understanding health equity and equity in the context of climate change separately and in cross-cutting domains; however, existence of a comprehensive framework for exploring the complexity of these interacting factors in health and climate change policies is hard to find. To date, most of the policy-analysis frameworks to understand heath equity in relation to climate change and public health has been in sector silos. Though there are also generic policy analysis frameworks that have been used to understand health equity in a multitude of policy domains including climate change, attempts have not been made to synthesize this evidence and such studies from LMICs are rare. Therefore, through this study, we aim to review existing heath-equity-focused policy-analysis frameworks and policy-analysis studies that have focused to understand health equity considerations in climate change policies and in health policies related to climate change (as shown in Figure 1). To our knowledge, this is the first attempt to track policy analysis frameworks and studies to understand health equity in climate change related policies. We anticipate that the findings from this review will contribute to the literature attempting to advance integrated research in health equity and climate change crisis and to center health equity in response to climate change related policies. The conceptual framework of the study is presented below.

### 1.2. Objectives

The primary objective of the review is to collate various policy analysis frameworks on health equity in climate change policies and in health policies related to climate change. The review will also examine, in published research, different policy analysis studies and summarize the approaches used to understand health equity in the intersection of climate change and public health policies.

### 1.3. Review Question

What policy analysis frameworks exist in the peer-reviewed literature to understand health equity in climate change policies and in public health policies related to climate change?

## 2. Materials and Methods

### 2.1. Protocol and Registration

The study has been registered in PROSPERO (CRD42021248734), the International Prospective Register of Systematic Reviews and will be carried out in accordance with the Preferred Reporting Items for Systematic Review and Meta-Analysis (PRISMA) guideline. This systematic review will be reported in accordance with the PRISMA-Equity 2012 Extension: Reporting Guidelines for Systematic Reviews with a Focus on Health Equity, and details of any changes to this protocol will be reported in the final review manuscript.

### 2.2. Eligibility Criteria

We will identify studies based on the PICOs criteria as outlined below. The study will consider all relevant studies in English language from year 2000 onwards (after the Millennium Development Goals were adopted in 2000, as this is one of the key milestones to address the linkage between climate change and public health) without any geographical restriction.

This review seeks to be a perspectival, rather than exhaustive, literature review; therefore, the search will be limited to English-language articles published from January 2000 onwards. Grey literature, books and chapters, commentaries, or opinion papers and editorials will not be included in the study. Any kind of reviews including systematic reviews will be excluded as we expect to cover most of the relevant literature through our searches, and a review of systematic reviews will be beyond the scope of this study.

#### 2.2.1. Inclusion

To meet the above-mentioned review objective, the specific criteria for inclusion are as follows:(1)Peer reviewed journal article that specifically reports on health equity frameworks used for analyzing climate change policies or public health policies related to climate change.(2)Published articles with a focus on policy analysis and health equity in the context of climate change policies.(3)Articles published from 2000 onwards and in English.

#### 2.2.2. Exclusion

(1)Generic Policy analysis (frameworks) that could be applicable in multiple policy domains (such as “health in All policies “, Universal Health Coverage, intersectionality-based policy analysis frameworks) unless the study is primarily about health equity in the context of climate change.(2)Analysis of policies domains with indirect linkage to climate change (such as housing, nutrition, healthy living, land use planning, insurance schemes, etc.), unless the study is tied to climate change policies or health policies related to climate change.(3)Assessments studies and frameworks assessing or evaluating the impacts of climate change policies or climate change related health policies on health equity outcomes.(4)Studies not published as a peer-reviewed article.

### 2.3. PICO

The PICO criteria for the study are as follows.

#### 2.3.1. Population

Studies/frameworks on analyzing health equity in the climate related health policies and climate change policies in both low- and middle-income countries and high-income countries.

#### 2.3.2. Intervention

Any articles explicitly either on ‘health equity focused policy analysis” or implicating health equity-related theories/frameworks/principles for specifically analyzing climate change policies or climate change related health policies.

#### 2.3.3. Comparison

Due to the nature of this review, this item for the PRISMA-E guideline will not be relevant.

#### 2.3.4. Outcomes

The primary and secondary outcomes of this review are as follows:

In climate change policies or health policies related to climate change,
(1)Different frameworks of policy analysis are used to understand health equity (primary outcome).(2)Different approaches/methods are undertaken to conduct policy analysis to understand health equity (secondary outcome).

### 2.4. Search Strategy

#### Databases

We will search five electronic databases (both public health and climate change specific as well as multidisciplinary databases covering international development). 

For published studies, the following databases will be searched:Public Health Specific: PubMed, CINAHL.Climate Change/environmental specific: GreenFILE.Cross Disciplinary: Web of Science Core collection and Wiley Online Library.

An initial literature search was done using key terms such as “Health Equity”, “climate change”, “Public Health”, “Environment and Public Health” and “Policy Making” with their synonyms. Following this, an initial scoping review was conducted in PubMed to identify appropriate keywords (and MeSH terms) and develop search strings. Search terms were disaggregated using truncation (‘*’) and wildcards as appropriate and key terms combined using Boolean operator terms AND and OR. Detailed search strings for all five databases were developed in consultation with a librarian at the University of Bielefeld. The search strings supporting the study are included within the article and provided as an additional file (S2).

### 2.5. Study Records and Management

#### 2.5.1. Data Management

Records from scientific databases will be imported into EndNote. The results from different databases will be combined and duplicate citations removed [29].

#### 2.5.2. Study Selection

The study will be screened for inclusion in two stages: stage 1, titles and/or abstracts; stage 2, full text. An initial screening of the title and abstract will be conducted to remove nonpertinent studies by two independent reviewers (SK, LR) based on predetermined criteria. In cases of ambiguity or disagreement, a decision of eligibility for inclusion will be made after consultation with the third reviewer (MB).

The full text of selected citations will also be assessed in detail against the inclusion criteria by two reviewers. During the full-text review, any studies found to not match the inclusion criteria will be removed and log of the excluded studies will be kept stating the reason for exclusion. Any difference in opinion will be solved through an agreement between the reviewers and in case of disagreement through discussion with a third reviewer (MB).

#### 2.5.3. Critical Appraisal

We will not appraise the quality of studies describing frameworks as we assume these to be largely descriptive. Because the nature of articles expected out of this review is diverse, critical appraisal will be conducted only for empirical research, where deemed appropriate.

For qualitative evidence synthesis: We will use the Critical Appraisal Skills Programme 2016 (CASP) [30]. The CASP checklist consist of 10 questions answered with a ‘yes’, ‘no’ or ‘cannot tell’. This includes two screening questions on aims of study and appropriateness of qualitative methodology and eight appraisal questions on research design, recruitment strategy, data collection, reflexivity-related issues, ethical issues, rigor of data analysis, and the reporting and value of findings.

For quantitative evidence synthesis, we will use the Strengthening the Reporting of Observational Studies in Epidemiology (STROBE) checklist. The checklist consists of 22 items. Eighteen items are common to all the three observational designs; that is, cohort, cross-sectional, and case–control studies and the remaining four checklists have specific variations according to the study design [31].

For Randomised Control Trials: We will use the CONSORT checklist to assess the quality of Randomised control Trials. The checklist items focus on reporting how the trial is designed, analysed, and interpreted; the flow diagram displays the progress of all participants through the trial [32].

The quality assessment of papers will be carried out by SK and the co-author (LR) and the steps will be validated by the third reviewer (MB).

## 3. Results

### 3.1. Data Extraction

We will employ a thematic analysis approach for analysis and extraction of data. Data extraction procedures will follow PRISMA guidelines. A codebook for data extraction will be developed iteratively. The descriptive themes included in the data extraction template will be author details, year of publication, study title, study aim and objectives, and study focus on relation to the policy cycle. For both qualitative and quantitative studies, data will be extracted and categorized according to country classification by income in 2020, study design, data collection method, type of analysis and policy (topic) field. For classification of countries, we will use the most up-to-date World Bank country classifications by income 2021–2022 [33] that include low, lower middle and upper middle-income countries.

Further, from each of the included publications, we will extract the frameworks used, policy analysis methods adopted, any definitions, theories, underlying principles explained for understanding health equity and record the frequency of each item across publications.

A data extraction form will be developed, and data extraction and quality assessment will be carried out by SK and 30% of the studies will be reviewed by a second author (LR). Any discrepancies on selection and extraction will be resolved through discussion.

### 3.2. Data Synthesis

Considering that the aim of this study is to identify various frameworks of policy analysis, the results will be discussed in narrative form. The frameworks identified will be grouped according to their policy field, country (HIC and LMICs) and assessed individually to synthesize the following features [34]:(a)type of the framework—descriptive, prescriptive, explanatory, or predictive(b)how health equity has been defined in terms of climate change(c)domains addressed in the framework for understanding health equity(d)comprehensiveness/depth/method of analysis of health equity(e)the context in which the framework is implemented

For any other policy analysis papers identified, in addition to the above (as relevant), the following information will be analyzed:(a)What methods have been used to analyze policies to understand health equity in the face of climate change.(b)Dimensions (/components) of health equity used in conducting policy analysis.

Conclusions will be drawn and adequacy of description of the definitions, concepts, and frameworks for policy analysis across the groupings summarized narratively.

### 3.3. Data Display 

To ensure transparency and replicability of the study processes, the data extracted will be described using a PRISMA Flow Diagram.

## 4. Conclusions

The study will provide insight into different policy-analysis frameworks that could be used to better understand health equity aspects more holistically in the climate-change-related policies. Knowledge on different frameworks will also be helpful to improve our understanding of the extent to which health equity is considered in the climate-change-related policies and contribute to inform the development of recommendations for research and practice.

## Figures and Tables

**Figure 1 ijerph-19-09126-f001:**
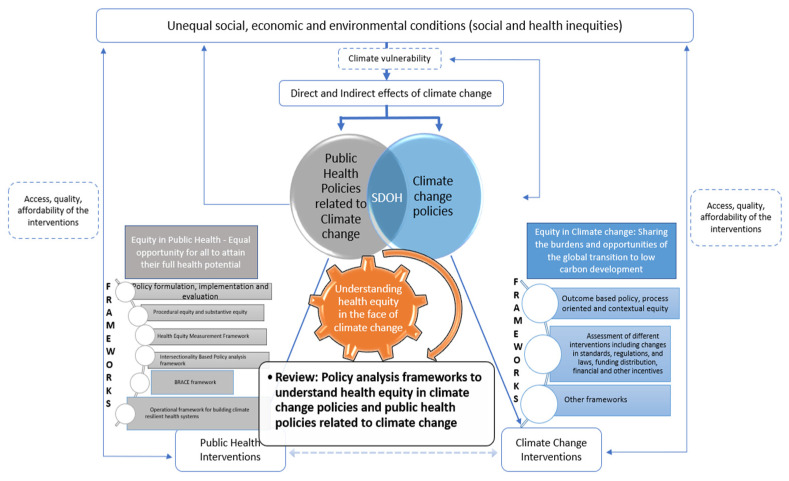
Conceptual framework of equity in health and equity in climate change and the review questions.

## Data Availability

Data sharing is not applicable to this article as no datasets were generated or analyzed during the current study. This is an ongoing study, and the data is contained within the article or supplementary material. The study will make all the data publicly available once the study is completed.

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
