# Peer review of "Health Equity in Climate Change Policies and Public Health Policies Related to Climate Change: Protocol for a Systematic Review"

_ijerph, 2022, doi:10.3390/ijerph19159126_

Round 1

Reviewer 1 Report

There is no doubt that the topic of this study is of great importance, and the researchers are commended for initiating this work. I have minor, although important, comments for the authors.

The Review Question: What policy analysis frameworks exist in the peer-reviewed literature to understand health equity in climate change policies and in public health policies related to climate change? may not get answered by looking at peer review literature alone. While the focus on peer review literature is understandable, in addressing polices there are important reports that are not published in journals, but which countries use on a regular basis, and are relevant to the topic. Several of these are from the World Health Organization, such as the Operational Framework for building climate resilient health systems; a document on Health in the National Action Plans (of the UNFCCC); and the Assessment of vulnerability and adaptation, among others. Even if not included as part of the literature, they could be discussed in the introduction, where other reports and tools are discussed. 

The proposed conceptual framework for the study separates public health policies related to climate change, and climate change policies. The inclusion of different frameworks in each is not clear. For example, BRACE should be in the public health side (note also that this is cited as a CDC report, i.e., it is not in the published literature.). The health equity measurement framework is listed in the public health policies related to climate change side, and yet, the paper cited does not even mention climate change. Some of the frameworks listed are likely not frameworks, such as the assessment of different interventions. The question that remains is, how to address global policy issues such as the Paris agreement (which has been considered to be also a global health agreement).

The comments above are also relevant for the Materials and Methods section.

Reviewer 2 Report

This is a timely and interesting study.

In the section of "For quantitative evidence synthesis", only the STROBE checklist is listed in the manuscript. As a systematic review, which would involve quantitative analysis of the content of the literature, it is recommended that the statistical methods used to conduct the literature analysis be added, if possible, to enhance the scientific validity of the protocol.
